# Direct Evidence on Effect of Oxygen Dissolution on Thermal and Electrical Conductivity of AlN Ceramics Using Al Solid-State NMR Analysis

**DOI:** 10.3390/ma15228125

**Published:** 2022-11-16

**Authors:** Jaegyeom Kim, Jong-Young Kim, Heewon Ahn, Mu Hyeok Jeong, Eunsil Lee, Keonhee Cho, Sung-Min Lee, Wooyoung Shim, Jae-Hwan Pee

**Affiliations:** 1Icheon Branch, Korea Institute of Ceramic Engineering and Technology (KICET), 3321, Gyeongchung Rd., Sindun-Myeon, Icheon-si 467-843, Gyeonggi-do, Republic of Korea; 2Department of Materials Sciences & Engineering, Multiscale Materials Laboratory, Yonsei University, 50, Yonsei-ro, Seodaemun-gu, Seoul 37022, Republic of Korea

**Keywords:** thermal dissipation, aluminum nitride, Al solid-state NMR, thermal conductivity, electrical conductivity, phonon scattering, aluminum vacancy, carbothermal

## Abstract

Aluminum nitride, with its high thermal conductivity and insulating properties, is a promising candidate as a thermal dissipation material in optoelectronics and high-power logic devices. In this work, we have shown that the thermal conductivity and electrical resistivity of AlN ceramics are primarily governed by ionic defects created by oxygen dissolved in AlN grains, which are directly probed using ^27^Al NMR spectroscopy. We find that a 4-coordinated AlN_3_O defect (O_N_) in the AlN lattice is changed to intermediate AlNO_3_, and further to 6-coordinated AlO_6_ with decreasing oxygen concentration. As the aluminum vacancy (V_Al_) defect, which is detrimental to thermal conductivity, is removed, the overall thermal conductivity is improved from 120 to 160 W/mK because of the relatively minor effect of the AlO_6_ defect on thermal conductivity. With the same total oxygen content, as the AlN_3_O defect concentration decreases, thermal conductivity increases. The electrical resistivity of our AlN ceramics also increases with the removal of oxygen because the major ionic carrier is V_Al_. Our results show that to enhance the thermal conductivity and electrical resistivity of AlN ceramics, the dissolved oxygen in AlN grains should be removed first. This understanding of the local structure of Al-related defects enables us to design new thermal dissipation materials.

## 1. Introduction

Thermal dissipation is a pre-requisite for the efficient and reliable operation of electronic devices from low-power logic devices to high-power RF devices. To achieve thermal mitigation and heat removal from hot spots, the use of high thermal conductivity materials with insulating electronic conductivity is ideal [1,2,3,4,5]. A few 3-dimensional crystals (diamond, c-BN, BAs, AlN, and GaN) and 2D-materials (graphene and h-BN) satisfy such requirements; however, thin films or polycrystalline materials inevitably lose the perfectness of their crystal quality to ensure ideal “in state of art” thermal conductivity. In addition, heterogeneous integration and the shrinkage of length scale, when materials (e.g., diamond) are implemented as heat spreaders into devices, lead to reduced conductivity [6,7,8,9]. Compared to the III-V semiconductor candidates, aluminum nitride (AlN) has good insulating properties and low thermal expansion coefficient (4.2 ppm @RT), with a theoretical thermal conductivity of 320 W/mK, making it a promising thermal dissipation material in optoelectronics, high-power RF devices, electrostatic chucks and heaters in CVD, and etchers [10,11,12]. Most electrically insulating ceramics (sapphire, SiO_2_) have low thermal conductivity because the thermal diffusivity of ceramics, in general, inversely scales with the electronic band gap; however, AlN is one of the rare examples of an electrically insulating ceramic with large phonon conductivity and a wide band gap (e.g., diamond, h-BN) [13,14,15]. Recently, bulk-like cross-plane thermal conductivity of micrometer-thick AlN film was achieved [16]. For the applicability of polycrystalline ceramics to heat-dissipative substrates for high-power RF devices, phonon scattering due to point defects in the AlN lattice, as well as boundary scattering due to the micrometer-sized grain boundary, should be controlled. Furthermore, ionic and electronic carriers, created via oxygen dissolution, should be reduced to maintain the insulation properties of AlN ceramics. Our objective in this study is to minimize the detrimental effect of point defects due to dissolved oxygen on the thermal and electrical conductivity of AlN ceramics.

Typical sintered AlN shows thermal conductivity, ranging from 150–190 W/mK. The key parameter determining phonon conduction is known to be oxygen dissolved in the lattice of AlN. Commercial AlN powder contains 0.8~1.0 wt% oxygen in general [17,18,19,20,21]. According to a previous study, the coordination number of oxygen in AlN changes critically at a concentration of 0.75 at% [22]. At lower concentrations, the well-known vacancy creation model by Slack et al. is applicable (e.g., 3O_N_ + V_Al_), in which the aluminum vacancy scatters phonons in the AlN lattice [11,12]. To explain the behavior in a higher oxygen concentration region, a defect-cluster model of octahedral-coordinated Al-O with an annihilation of 4-coordinated Al-O defect is proposed by Harris et al., which is indirectly suggested by luminescence and a change in lattice parameter [22]. The proposed Al-O defect would scatter phonons due to a lattice mismatch, stacking faults, and boundary effects, subsequently reducing thermal conductivity. The understanding of the local structure and chemical bonding of such defect clusters in an AlN lattice is critically important for the improvement of thermal conductivity.

In this work, we attempt to systematically study the effect of oxygen dissolution on the thermal and electrical conductivity of AlN via Al solid-state NMR analysis. The local coordination of aluminum is directly probed using NMR analysis, showing that 4-coordinated AlN_3_O defect clusters (O_N_) are present in raw materials. The 4-coordinated cluster is changed to AlNO_3_ of the intermediate γ-ALON phase, and further to 6-coordinated AlO_6_ with decreasing oxygen concentration. The resulting thermal conductivity increases by up to ~33% (160 W/mK) as a result of the removal of oxygen when dissolved in AlN grains. Furthermore, the resulting electrical resistivity increases with decreasing oxygen concentration, which also supports reduced ionic carriers (V_Al_) created by oxygen dissolution. Our results show that the 4-coordinated Al-O defects should be removed from AlN grains to attain high thermal conductivity and electrical resistivity simultaneously, which would be beneficial to the enhanced heat-dissipative substrate and electrostatic chucking dielectrics.

## 2. Materials and Methods

AlN raw materials with controlled oxygen concentration were synthesized via a carbothermal reaction between Al_2_O_3_ and carbon black. High-purity Al_2_O_3_ powder and carbon black powder were purchased from Chalco Shandong Co., ltd. and the Columbian Chemicals Company, respectively. Carbon black materials of different sizes (D50 = 0.014, 0.045 μm) were use as reactants, which were denoted as C-S and C-L, respectively. The Al_2_O_3_ and carbon black were uniformly mixed in a molar ratio of 1:3.1 via wet ball milling using anhydrous ethanol. The resultant solution of Al_2_O_3_ and carbon black was then completely dried in a drying oven at 150 °C. The prepared mixture (~20 g) was placed into a graphite crucible, and then, heated at 1700 °C for 1 to 6 h in a homemade graphite furnace. The carbothermal process was performed under high-purity N_2_ gas at a flow rate of 3 L/min. Subsequently, the powder was heated in air at 700 °C for 1 h to remove the excess carbon. Commercial AlN powder, purchased from Tokuyama Corp. (Chiyoda, Japan), was used as a reference material. Sintered specimens were obtained via pressureless sintering of a mixture of AlN and Y_2_O_3_ (4 wt%) at 1850 °C for 4 h in N_2_ atmosphere.

The densities of the samples were measured using the Archimedes method. The oxygen content of the AlN powder was measured using an oxygen/nitrogen analyzer (EMGA-920, HORIBA, Kyoto, Japan) for more than 3 samples. The microstructures were observed using field emission scanning electron microscopy (FE-SEM; JSM-6710F, JEOL). The crystal structures of the major and minor phases in the samples were investigated using an X-ray diffractometer (D/max-2500, RIGAKU, Tokyo, Japan). Thermal diffusivity was measured using an XFA500 (Linseis). Heat capacity was assumed to be 0.73 Jg/K. To measure the volume electrical resistivity, a platinum electrode was deposited on polished samples via sputtering. High-voltage electrical resistivity was measured using a high-resistance meter (Keysight B2985A, Santa Rosa, CA, USA) at 25~550 °C after 60 s, applying a voltage of 100 V/mm. ^27^Al solid-state NMR spectra were obtained using a 600 MHz high-resolution FT-NMR spectrometer (Varian INOVA 600 FT NMR; Varian Inc., Palo Alto, CA, USA).

## 3. Results

### 3.1. Synthesis

The AlN raw materials with controlled oxygen concentration were synthesized by controlling the carbothermal reaction time (Figure 1). The powder mixture of alumina and carbon was reacted at 1700 °C for 1 to 6 h. The XRD patterns of the AlN samples after the reaction are shown in Figure 2. The diffraction patterns show that monophasic AlN with a wurtzite-type structure was synthesized after reaction for more than 1 h. Peaks of Al_2_O_3_ remain in the XRD pattern for 1 h, owing to the incomplete conversion of Al_2_O_3_ to AlN. This indicates that reaction at 1700 °C for more than 1 h was required to convert α-Al_2_O_3_ into the wurtzite-type AlN. With the same size of Al_2_O_3_, when smaller carbon with a particle size of 0.15 μm was used (AlN_C-S), the remaining oxygen was smaller, probably because of the more intimate contact between Al_2_O_3_ and the carbon particles, as shown in Table 1. In the XRD patterns in Figure 2, with increasing reaction time for the sintered AlN_C-S/C-L materials, peaks due to the second phase of Y_2_Al_4_O_9_(YAG) change to those due to Y_4_Al_2_O_9_(YAM) and YAlO_3_(YAP), whereas only peaks due to the YAP phase are observed for commercial materials (Appendix A). As the ratio of Y_2_O_3_ (from the sintering aid) to Al_2_O_3_ (from the remaining oxygen in AlN) increases, the formation of YAM is preferred to YAP and further to YAG, possibly because Y/Al for YAM is larger than YAP and YAG. Therefore, when there is more remaining oxygen, (AlN_C-L), the YAG phase is observed in the XRD patterns, even after carbothermal reaction for 6 h.

### 3.2. ^27^Al Solid-State NMR

#### 3.2.1. Raw Materials

The ^27^Al solid-state NMR spectra of the AlN raw materials are shown in Figure 3. AlN_C-S and AlN_C-L were chosen as raw materials, and commercial AlN powder was used as a reference. In the NMR spectra of raw materials (Figure 3a), main peaks at 120 ppm due to AlN_4_ tetrahedra are observed, and shoulders at ~110 ppm are also found. As carbothermal time increases from 2 h to 6 h for the AlN_C-S and AlN_C-L materials, the intensity of the shoulder at ~110 ppm, which corresponds to 4-coordinated Al (AlN_3_O tetrahedra), decreases, as shown in Figure 3c [23,24]. Around 10 ppm, small peaks due to 6-coordinated Al defect (AlO_6_) are found for our materials and the commercial materials, as shown in Figure 3e. The AlO_6_ sites are probably present in the grain boundaries or surfaces of the AlN grains in small amounts, and therefore, XRD peaks due to Al_2_O_3_ are absent or very weak for the raw materials (Figure 2). As shown in Figure 3d, commercial materials exhibits almost no shoulder peaks (~110 ppm) compared to AlN_C-S, which means most of the remaining oxygen is not present in the AlN grains. Even though the oxygen content of the commercial material is almost the same as that in AlN_C-S (5/6 h) (Table 1), the amount of oxygen dissolved in the AlN grain is smaller for the commercial materials, which is shown by the AlN_3_O NMR peak area (4.73% and 0% for AlN_C-S (5 h) and commercial, respectively).

#### 3.2.2. Sintered Samples

The ^27^Al solid-state NMR spectra of the sintered AlN samples are shown in Figure 4. In the NMR spectra of the sintered AlN_C-S samples, the peaks (shoulder) at ~110 ppm, excluding the main peak (~120 ppm), a significantly weakened by sintering, which means that the defect due to oxygen substitution in the AlN lattice (i.e., O_N_) was removed. Instead, new peaks due to the Al-O defects are found in the spectra of the sintered samples. The new peaks at 0 and 60–70 ppm correspond to the 4-coordinated AlNO_3_ of a probable γ-AlON phase, whereas the peaks at ~10 ppm correspond to the 6-coordinated AlO_6_ cluster, like those in the spectra of the raw materials [23,24]. As the carbothermal reaction time increases from 2 h to 6 h for AlN_C-S materials, peaks due to the γ-AlON phase gradually decrease, whereas peaks at 10 ppm due to the AlO_6_ octahedra increase, as shown in Figure 3. As carbothermal time increases, the intensity of the shoulder at ~110 ppm also decreases for the sintered samples, which corresponds to 4-coordinated Al (AlN_3_O).

### 3.3. Thermal Conductivity and Electrical Resistivity

The thermal conductivity of the sintered samples of AlN_C-S/AlN_C-L and of the the commercial material was measured at room temperature. As shown in Figure 5a, thermal conductivity increases with increasing carbothermal reaction time, which shows that thermal diffusivity is inversely proportional to the concentration of oxygen dissolved in the AlN grains. The oxygen in the AlN grains, which is actually in the form of AlN_3_O, is diffused out during the sintering process via intermediate AlNO_3_, as shown in the NMR analysis. The reaction of the additive (Y_2_O_3_) with dissolved oxygen, when these two are in physical contact at the grain boundary, leads to the second phases of YAP or YAM or of the oxide layer at the grain boundary. The overall oxygen gathering process may occur via dissolution and reprecipitation, and therefore, the grain size of the raw materials affects the kinetics of the mass transport reaction. Because the grain size of AlN_C-S is smaller than that of AlN_C-L, the thermal conductivity of the sintered AlN_C-S increases more than that of AlN_C-L with increasing carbothermal reaction time. Even though the oxygen concentration of AlN_C-L is slightly larger than that of AlN_C-S, the thermal conductivity exhibits significantly reduced values, probably because of the larger grain size of C-L (see particle size, BET, and SEM image of AlN_C-S/C-L in Appendix A). As shown in Figure 5b, the electrical resistivity of the AlN_C-S materials increases with increasing carbothermal reaction time from 2 h to 5 h. The mechanism of increase in electrical resistivity seems to be pertinent to that of the increase in thermal conductivity, which will be discussed in a later section. The sintered commercial material exhibits higher thermal conductivity (160 W/mK) than the AlN_C-S materials.

## 4. Discussion

Slack proposed that the oxygen dissolved in the AlN lattice produces defects via an ionic compensation reaction, according to Equation (1) [11]. An aluminum vacancy is created for every three oxygen atoms accommodated in the AlN lattice, which effectively lowers the thermal conductivity.
(1)$${\mathrm{Al}}_{2}O_{3}\overset{3\mathrm{AlN}}{\to }2{\mathrm{Al}}_{\mathrm{Al}}^{x}+3O_{N}^{\cdot }+V_{\mathrm{Al}}^{\prime \prime \prime }$$

The phonon scattering cross-section (Γ) is more influenced by vacancy formation (Al → V_Al_) than by oxygen substitution (Al-N → Al-O) because the atomic weight difference between Al and the vacancy (Δ*M*/*M* = 1.0) is larger than that between N and O (Δ*M*/*M* = 0.14) [25,26].
(2)$$\Gamma =X_{S}\left(1-X_{S}\right)\left[{\left(\frac{\Delta M}{M}\right)}^{2}+\varepsilon {\left(\frac{\Delta \delta }{\delta }\right)}^{2}\right]$$
where Δ *M*/*M* and Δδ/δ are the mass and strain misfits, respectively, *ε* is a dimensionless parameter, and X*s* is the solute concentration. Therefore, Al vacancy formation is greatly detrimental to phonon thermal conductivity when the oxygen concentration is smaller than 0.75 at% (~3 × 10^20^/cm^3^). For example, the oxygen concentration is increased by a factor of 10, and the thermal conductivity reduction can be roughly estimated by a factor of √10 since thermal conductivity scales with the scattering cross-section, κ ∝ Γ^−0.5^, at intermediate temperature [25]. These estimations are supported by recent work via the first-principle calculation [27]. In the low-oxygen-concentration regime, therefore, the most conductivity-determining defect is expected to be an AlN_3_O tetrahedral cluster.

However, in most cases of polycrystalline ceramic fabrication, the oxygen concentration often exceeds 0.8 wt% (2.0 at%, ~10^21^/cm^3^). According to Harris et al., a 6-coordinated AlO_6_ defect is formed via the annihilation of an Al vacancy defect, as the oxygen concentration exceeds the critical concentration (>0.75 at%). In this regime of high oxygen concentration, the dissolved oxygen would be present as an AlO_6_ defect cluster in the second phases, or as an oxide layer at the grain boundary, which can potentially scatter phonon transport. However, the oxygen-containing second phases have a relatively minor effect on thermal conductivity because Δ*M*/*M* = 0.14 upon substitution of oxygen in a nitrogen site, unless the AlN grains are noncontiguous.

In our work, the oxygen concentration of the AlN_C-S raw materials was reduced by ~43% with increasing carbothermal reaction time (from 2 h to 6 h) according to NO elemental analysis. Accordingly, the 4-coordinated Al(N,O)_4_ cluster is expected to be removed, which is very consistent with the reduced NMR peak area due to AlN_3_O (~58%) for the samples. The commercial material exhibits an almost completely removed AlN_3_O defect according to the NMR peak area calculation (Table 1). The presence of AlN_3_O peaks in the NMR spectra of the raw materials directly indicates the formation of O_N_ and V_Al_ via the incorporation of oxygen into the AlN lattice. The decreased shoulder peak at 110 ppm in the Al NMR spectra of the sintered samples provides evidence that such a defect was removed, in part, by sintering, as shown in Table 1. The experimentally observed increase in thermal conductivity is ~33% (120 W/mK vs. 160 W/mK for AlN-C-S (2 h) and commercial material, respectively) as a result of the removal of AlN_3_O, which is much smaller than the theoretical value (320 W/mK); however, the additional effect of the newly formed 6-coordinated defects should be considered. Despite the relative minor effect on the thermal conductivity of AlO_6_ defects, phonon scattering by the oxygen-related second phase is still present. Furthermore, according to the first-principle calculation, as defect concentration decreases, improvements in thermal conductivity are shown to be limited owing to the size effect [27].

When compared to commercial materials, even though the oxygen concentration values in our materials are almost the same according to the chemical analysis (Table 1), thermal conductivity of the commercial material is still higher. This result might be attributed to lower oxygen concentration in the AlN grains (i.e., the AlN_3_O defect) of the commercial material than AlN_C-S/C-L, as evidenced by the Al NMR analysis. The V_Al_ defect, created via oxygen dissolution in the AlN grains, is crucial to phonon scattering, and therefore, the oxygen in the AlN grains should be removed first to enhance thermal conductivity.

As shown in Figure 5b, the high-voltage electrical resistivity of AlN_C-S materials increases with increasing carbothermal reaction time from 2 h to 6 h. This result is also very consistent with the removal of O_N_ and V_Al_, created via the incorporation of oxygen into the AlN lattice, because the ionic conductivity of AlN ceramics is governed by vacancy in the carrier of aluminum (V_Al_) [28,29,30,31,32]. Such enhanced thermal conductivity and electrical resistivity offer excellent characteristics for use as thermal dissipation materials and electrostatic chucking dielectrics.

## 5. Conclusions

In this work, we attempted to elucidate the oxygen dissolution effect on the thermal conductivity and electrical resistivity of AlN ceramics. According to the carbothermal reaction condition, oxygen impurity was found to be dissolved in a wurtzite-type AlN lattice with tetrahedral symmetry or at the grain boundary/second phase with octahedral symmetry. ^27^Al NMR spectroscopic analysis clarifies that 4-coordianated AlN_3_O defects are present in the raw materials along with the 6-coordinated AlO_6_ defect. As the carbothermal reaction proceeds, the AlN_3_O tetrahedra, which provide evidence of the presence of O_N_, decrease. The oxygen in the raw materials turns into that of γ-AlON phase, and further into that of second phase with octahedral local symmetry, such as YAP, as a result of sintering. As aluminum vacancy (V_Al_), which is crucial to phonon scattering, decreases with decreasing oxygen content, thermal conductivity increases from 120 W/mK to 160 W/mK. Comparison with commercial materials without AlN_3_O defects indicates that the presence of V_Al_ (O_N_) is critical to phonon scattering compared to AlO_6_. High-voltage electrical resistivity also increases with decreasing oxygen content, which shows a detrimental effect of V_Al_ on resistivity. The obtained results suggest a vacancy mechanism, created by oxygen impurity in AlN ceramics, which makes it easy to design thermal dissipative materials for high-power logic devices and highly insulating electrostatic chucking dielectrics for semiconductor manufacturing equipment.

## Figures and Tables

**Figure 1 materials-15-08125-f001:**
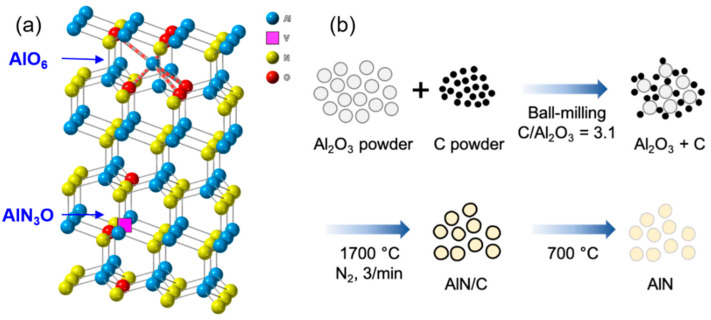
(**a**) Crystal structure of wurtzite-type AlN. 4-coordinated AlN_3_O defect, involving aluminum vacancy (V_Al_), and 6-coordinated AlO_6_ defect are shown. (**b**) Synthesis of aluminum nitride via carbothermal reaction. Oxygen concentrations of AlN materials were controlled by carbothermal reaction time at 1700 °C. Commercial material from Tokuyama corp. was used as a reference. SEM images of alumina, carbon(C-S/C-L), and aluminum nitride (AlN_C-S, AlN_C-L) are shown in Appendix A.

**Figure 2 materials-15-08125-f002:**
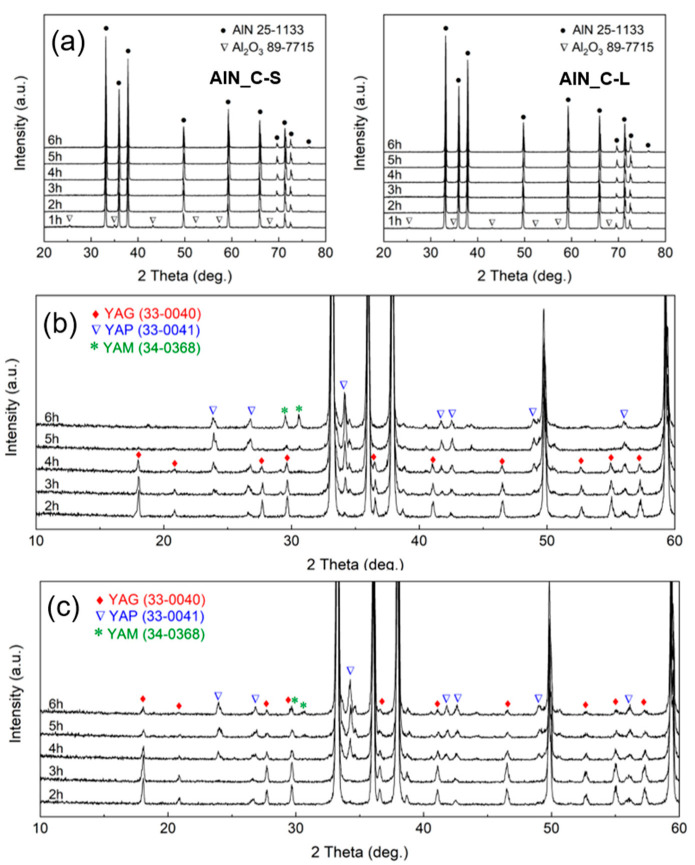
XRD patterns for (**a**) raw materials (AlN_C-S, AlN_C-L), (**b**) sintered AlN_C-S samples, and (**c**) sintered AlN_C-L samples. As carbothermal reaction exceeds 1 h, peaks due to aluminum oxide in raw materials disappear as a result of reduced oxygen. As carbothermal reaction time increases, a second phase of YAG changes to YAP and YAM in the XRD patterns of sintered samples.

**Figure 3 materials-15-08125-f003:**
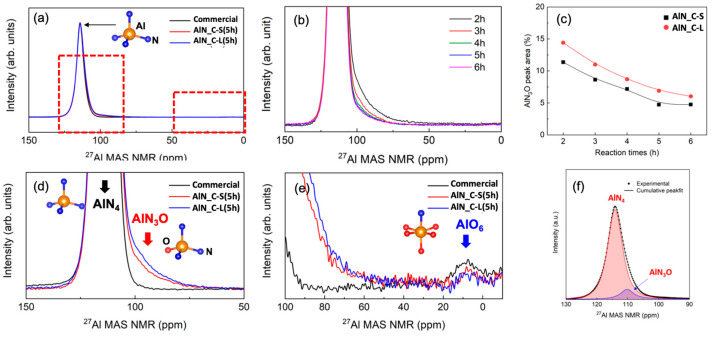
(**a**) ^27^Al solid-state NMR spectra for raw materials (AlN_C-S/C-L) showing shoulder peaks (~110 ppm) due to AlN_4_O defects. Very weak peaks due to AlO_6_ defects are found for AlN_C-S/C-L and commercial material (see Figure 3e for enlarged image). (**b**) The evolution of shoulder peak at ~110 ppm for AlN_C-S as a function of carbothermal reaction time. The intensity of the peaks decreases as oxygen concentration decreases with increasing carbothermal reaction time. (**c**) Fitted peak areas due to AlN_3_O defects of the main peak in Al NMR spectra are shown. (**d**) Comparison of AlN_3_O peaks of AlN_C-S/C-L (5 h) with commercial materials. (**e**) Comparison of AlO_6_ peaks of AlN_C-S/C-L (5 h) with commercial materials. (**f**) Fitting of experimental spectra with AlN_3_O and AlN_4_ profiles.

**Figure 4 materials-15-08125-f004:**
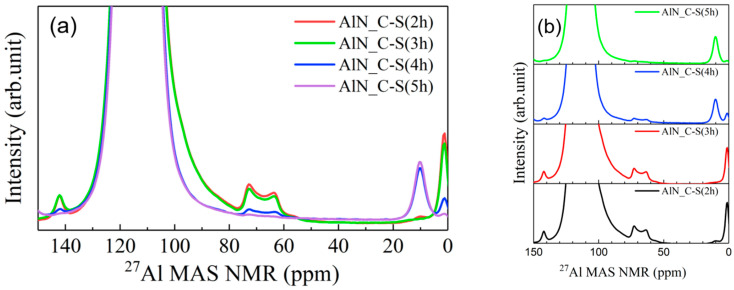
(**a**) Al solid-state NMR spectra for sintered samples (AlN_C-S materials with carbothermal reaction time of 2–5 h). After sintering, the peaks due to AlN_3_O defect (~110 ppm) near main peak (AlN_4_) were removed, as shown in Table 1. (**b**) As carbothermal reaction time increases (decreasing oxygen concentration), peaks due to AlNO_3_ defect of intermediate γ-AlON (60–70 ppm) decrease with increasing peak intensity due to AlO_6_ defect (~10 ppm).

**Figure 5 materials-15-08125-f005:**
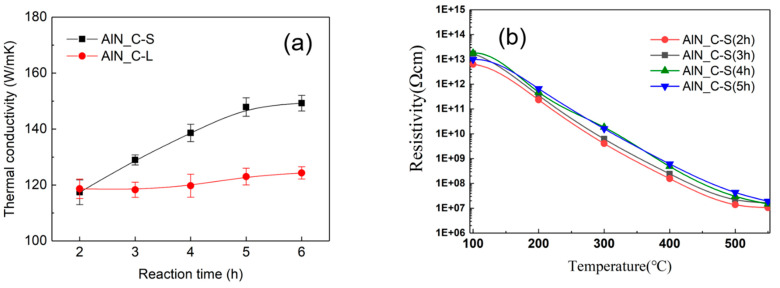
(**a**) Thermal conductivity of sintered AlN_C-S and AlN_C-L materials. AlN_C-S with smaller oxygen level and grain size exhibits more thermal conductivity than AlN_C-L. However, commercial materials without 4-coordinated AlN_3_O sites shows more thermal conductivity (160 W/mK) than AlN_C-S despite having almost the same oxygen content. (**b**) High-voltage electrical resistivity of sintered samples of AlN_C-S materials (100 V/mm). As carbothermal reaction time increases, electrical resistivity increases as a result of reduced carrier, aluminum vacancy (V_Al_).

**Table 1 materials-15-08125-t001:** Chemical analysis and NMR fitting results of AlN raw materials.

Reaction Time	O (wt%/at%)		C (ppm)		AlN_3_O NMR Peak Area (Raw Material, %)		AlN_3_O NMR Peak Area (Sintered, %)
	AlN_C-S	AlN_C-L	AlN_C-S	AlN_C-L	AlN_C-S	AlN_C-L	AlN_C-S
2 h	1.47/3.77	1.57/4.02	1691	2761	11.37	14.42	10.27
3 h	1.22/3.15	1.31/3.36	1527	2416	8.63	11.01	9.86
4 h	1.03/2.64	1.16/2.97	1412	2188	7.17	8.7	4.59
5 h	0.86/2.20	0.99/2.54	1087	2097	4.72	6.9	2.91
6 h	0.84/2.15	1.01/2.59	1132	2072	4.75	6.04	4.29
Commercial	0.85/2.17		280		0.0		-

## Data Availability

The data presented in this study are available from the corresponding authors upon reasonable request.

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
