# Peer review of "Direct Evidence on Effect of Oxygen Dissolution on Thermal and Electrical Conductivity of AlN Ceramics Using Al Solid-State NMR Analysis"

_materials, 2022, doi:10.3390/ma15228125_

Round 1

Reviewer 1 Report

Paper needs to be improved in following manners, this is a nice study however, following

questions are necessary to be answered before further processing

a. This would be beneficial if authors could provide more details at the end of introduction

specifically stating the objective of the paper, although this is explained but needs a little more

clarity.

b. Authors need to update the survey of literature for more recent papers specifically published in

the recent years 2022 etc.

The authors should elaborate on their new findings that are worthy of consideration for publication in a journal.

c. Language of the paper needs professional touch ups as there are typos and errors in some parts

of paper and they need to be reduced.

d. In the conclusion section, authors need to focus on the outcomes of their study with salient

findings only, keep them brief, as more explanation is already added in the results and

discussion section.

e. Results and discussion section is well explained, please try to look at figures in this section they might need more explanation if needed.

f. Altogether after these improvements are properly made, paper would be in a decent shape and

can be considered for publication if revised well.

g. Please give a bird eye view picture of your finding in abstract.

Reviewer 2 Report

This manuscript studies the effect of oxygen dissolution on thermal- and electrical-conductivity of AlN by Al solid-state NMR analysis. The thermal dissipation and resistance characteristics of AlN ceramics were explained from the perspective of the microscopic local structure of defects. Some interesting results were found. This manuscript could be accepted for publication after a mandatory revision. The comments are as following:

1.  The authors explained the thermal dissipation and resistance properties of AlN ceramics from the perspective of the microscopic local structure of defects, and suggest that the concentration of dissolved oxygen affects the thermal conductivity and insulation properties of AlN ceramics. Please explain in detail how the change in the structure of local defects affects the properties from a "structure-property" perspective.

2.  Two different sizes of carbon materials were used in this work, but the effect of different sizes on dissolved oxygen concentration was not detailed. Also, will size effect of high-purity alumina influence dissolved oxygen concentration.

3.   When the carbon thermal reaction time was 6 hours, why the second phase YAG was only present in the raw material (AlN_C-L)?

4.  A common method used to analyze the conductivity mechanism of ceramics is AC impedance spectroscopy. Please analyze and compare electrical conductivity properties obtained by AC impedance spectroscopy and NMR.

5. There are some English grammar and format errors.

Reviewer 3 Report

Introduction could be extended using some of the suggested refrences:

1. Mickael Coëffe-Desvaux , Nicolas Pradeilles,Pascal Marchet ,Marion Vandenhende ,Mickael Joinet  and Alexandre Maître

Comparative Study on Electrical Conductivity of CeO2-Doped AlN Ceramics Sintered by Hot-Pressing and Spark Plasma Sintering, Materials 2022, 15(7), 2399; https://doi.org/10.3390/ma15072399

2.  A. Franco JúniorD. J Shanafield, Thermal conductivity of polycrystalline aluminum  nitride (AlN) ceramics,• Cerâmica 50 (315) • Sept 2004 • https://doi.org/10.1590/S0366-69132004000300012

3.Debabrata Gangulya,Abhijit Beraa, Roumita Horea, Sipra Khanraa ,Pradip KMajib, Dinesh Kumar Kotneesc, Santanu Chattopadhyaya, Coining the attributes of nano to micro dual hybrid silica-ceramic waste filler based green HNBR composites for triple percolation: Mechanical properties, thermal, and electrical conductivity, , Chemical Engineering Journal Advances, Volume 11, 15 August 2022, 100338, , https://doi.org/10.1016/j.ceja.2022.100338

Please, use the refrences in text, as suggested in Instructions for Authors. Also add some related references from Metals, (2022, 2021, 2020). State of the art published could be improved using more recent references.

Results

Figure 1, are they the original figures, or from the literature? If some reference is used, it have to be mentioned.

Discussion is missimg, please add to the manuscript.

Conclusion is missing

Round 2

Reviewer 3 Report

Refrences

Please, check citation of suggested refrences, it appears, they are not cited by surnamens of the authors.

1) Mickael Coëffe-Desvaux , Nicolas Pradeilles,Pascal Marchet ,Marion Vandenhende ,Mickael Joinet  and Alexandre Maître, Comparative Study on Electrical Conductivity of CeO2-Doped AlN Ceramics Sintered by Hot-Pressing and Spark Plasma Sintering, Materials 2022, 15(7), 2399; https://doi.org/10.3390/ma15072399

2) A. Franco JúniorD. J Shanafield, Thermal conductivity of polycrystalline aluminum nitride (AlN) ceramics, Cerâmica 50 (315) Sept 2004 https://doi.org/10.1590/S0366-69132004000300012

3) Debabrata Gangulya,Abhijit Beraa, Roumita Horea, Sipra Khanraa ,Pradip KMajib, Dinesh Kumar Kotneesc, Santanu Chattopadhyaya, Coining the attributes of nano to micro dual hybrid silica-ceramic waste filler based green HNBR composites for triple percolation: Mechanical properties, thermal, and electrical conductivity, , Chemical Engineering Journal Advances, Volume 11, 15 August 2022, 100338, , https://doi.org/10.1016/j.ceja.2022.100338

Conclusion

Our work elucidated the vacancy mechanism, created by oxygen impurity in AlN ceramics, which make it easy to design thermal dissipative materials for power logic devices and highly insulating electrostatic chucking dielectrics for semiconductor manufacture equipment.

it is better to change into

The obtained results relted to the ..... instead Our work

Author Response

Dear reviewer

Thank you for your comment. We revised the manuscript according to reviews comment.

Comment1 ) Please, check citation of suggested refrences, it appears, they are not cited by surnamens of the authors.

1) Mickael Coëffe-Desvaux , Nicolas Pradeilles,Pascal Marchet ,Marion Vandenhende ,Mickael Joinet  and Alexandre Maître, Comparative Study on Electrical Conductivity of CeO2-Doped AlN Ceramics Sintered by Hot-Pressing and Spark Plasma Sintering, Materials 2022, 15(7), 2399; https://doi.org/10.3390/ma15072399

2) A. Franco JúniorD. J Shanafield, Thermal conductivity of polycrystalline aluminum nitride (AlN) ceramics, Cerâmica 50 (315) Sept 2004 https://doi.org/10.1590/S0366-69132004000300012

3) Debabrata Gangulya,Abhijit Beraa, Roumita Horea, Sipra Khanraa ,Pradip KMajib, Dinesh Kumar Kotneesc, Santanu Chattopadhyaya, Coining the attributes of nano to micro dual hybrid silica-ceramic waste filler based green HNBR composites for triple percolation: Mechanical properties, thermal, and electrical conductivity, , Chemical Engineering Journal Advances, Volume 11, 15 August 2022, 100338, , https://doi.org/10.1016/j.ceja.2022.100338

Answer1)  We corrected author's name of references and added reference.  Please refer to revised manuscript.

Comment2)

Conclusion

Our work elucidated the vacancy mechanism, created by oxygen impurity in AlN ceramics, which make it easy to design thermal dissipative materials for power logic devices andhighly insulating electrostatic chucking dielectrics for semiconductor manufacture equipment.

it is better to change into

The obtained results relted to the ..... instead Our work

Answer2)  We corrected the expression in Conclusion part.  Please refer to revised manuscript.
